# Computational Interaction Analysis of *Sirex noctilio* Odorant-Binding Protein (SnocOBP7) Combined with Female Sex Pheromones and Symbiotic Fungal Volatiles

Yi-Ni Li [1,2,†], En-Hua Hao [1,2,†], Han Li [1,2], Xiao-Hui Yuan [3], Peng-Fei Lu [1,2,*] and Hai-Li Qiao [4,*]

1   Beijing Key Laboratory for Forest Pest Control, Beijing Forestry University, Beijing 100083, China; lynn2021@bjfu.edu.cn (Y.-N.L.); hao452115308@126.com (E.-H.H.); 18610268286@163.com (H.L.)
2   The Key Laboratory for Silviculture and Conservation of the Ministry of Education, School of Forestry, Beijing Forestry University, Beijing 100083, China
3   Institute of Biomedicine, Jinan University, Guangzhou 510632, China; yuanhui1024@gmail.com
4   Institute of Medicinal Plant Development, Chinese Academy of Medical Sciences and Peking Union Medical College, Beijing 100193, China
*   Correspondence: lpengfei224@126.com (P.-F.L.); qhl193314@sina.com (H.-L.Q.);
Tel.: +86-10-6233-6755 (P.-F.L.); +86-10-5783-3180 (H.-L.Q.)
†   Yi-Ni Li and En-Hua Hao contributed equally to this work.

**Abstract:** *Sirex noctilio*, a major forestry quarantine pest, has spread rapidly and caused serious harm. However, existing methods still need to be improved because its olfactory interaction mechanisms are poorly understood. In order to study the role of male-specific protein SnocOBP7 in the protein–ligand interactions, we selected it as the object of computational simulation and analysis. By docking it with 11 ligands and evaluating free binding energy decomposition, the three best binding ligands were found to be female sex pheromones ((Z)-7-heptacosene and (Z)-7-nonacosene) and symbiotic fungal volatiles ((−)-globulol). Binding mode analysis and computational alanine scanning suggested that five residues play key roles in the binding of each female sex pheromone to SnocOBP7, whereas two residues play key roles in (−)-globulol binding. Phe108 and Leu36 may be the crucial sites via which SnocOBP7 binds female sex pheromones, whereas Met40 may regulate the courtship behavior of males, and Leu61 may be related to mating and host finding. Our studies predicted the function of SnocOBP7 and found that the interaction between SnocOBP7 and pheromone is a complex process, and we successfully predicted its binding key amino-acid sites, providing a basis for the development of new prevention and control methods relying on female sex pheromones and symbiotic fungi.

**Keywords:** *Sirex noctilio* Fabricius; odorant-binding protein; molecular dynamics; quarantine pest; biological interaction; computational simulation

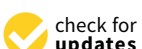



## 1. Introduction

As one of the main mechanisms through which insects perceive the external environment, the olfactory system plays a very important role in foraging, defense, mating, reproduction, information exchange, and habitat selection [1]. Studying how odor molecules in the environment act on insect sensilla and induce insects to produce behavioral responses will illuminate the molecular interaction mechanisms underlying insect host identification, interspecific interaction, and intraspecific communication, allowing the formulation of corresponding management and protection strategies with advantages over currently available methods.

Signal transduction in the peripheral olfactory system of insects can be summarized as follows: odor molecules enter the antennal sensilla lymph through the stratum corneum, where they bind to odorant-binding proteins (OBPs) or chemosensory proteins (CSPs) [2]. Soluble proteins transport hydrophobic odorants to membrane-bound odorant receptors (ORs) or ionotropic receptors (IRs) located on the dendritic membrane of olfactory neurons

through the sensory lymph, after which ORs or IRs convert chemical signals into electrical signals and transmit them to the central nervous system, thereby causing an olfactory response [3,4]. Finally, odor molecules are degraded by odor-degrading enzymes (ODEs) [5]. In general, after entering the organism through a series of olfactory proteins, the odor molecules are recognized and degraded, which is regarded as a vital ecological interaction process between organisms and the environment.

The combination of OBPs and odor molecules is the initial biochemical reaction through which insects specifically recognize external odorants [6]. A typical OBP has a stable and compact three-dimensional binding hydrophobic cavity, which is likely to be the key region involved in binding with external odor molecules [7,8]. However, OBPs are specific to the selection of environmental odor molecules based on their expression levels in different tissues. Correspondingly, through the regulation of the concentration or presence of certain odor molecules, organisms will affect the response of the biological olfactory system from the molecular perspective, and then manifest in the behavioral interactions. Thus, exploring how OBPs bind ligands in their binding cavity and revealing the binding interaction mechanisms underlying the effects of chemical volatiles on insects are important research subjects.

With the development of computer technology, computer simulations have been widely used in interdisciplinary studies of protein–ligand binding methods. Without getting the actual structure of the protein, homology modeling has become a commonly used method of predicting protein structure because proteins with high homology generally have similar three-dimensional structures [9]. For example, MvitOBP3 in *Maruca vitrata* was specifically predicted to bind to several host plant volatiles from legumes through homology modeling and molecular docking analysis [10]. However, because molecular docking only models transient and stable binding, the binding modes of protein and odorant molecules are not fully simulated. Therefore, molecular dynamics (MD) simulations must be included to determine whether the binding between a protein and odorant is stable, as well as to predict the major driving forces. For example, using a MD simulation, the seventh α-helix of *Agrilus mali* OBP8 quickly formed a loop structure upon binding with geranyl formate, which indicated that insect OBPs may be able to modify their secondary structures to increase the range of proteins with which they may bind [11]. In work aimed at deciphering the binding mechanism of *Athetis lepigone* GOBP2 to Chlorpyrifos and Phoxim, MD simulation was used to analyze the forces driving these interactions, revealing that the main driving forces were alkyl–π and hydrophobic interactions [12].

*Sirex noctilio* Fabricius (wood wasp; Hymenoptera: Siricidae: *Sirex*) is a major international forestry quarantine pest that is native to Eurasia and North Africa and has a wide range of hosts, mainly *Pinus*, and a few species in *Picea*, *Abies*, and *Larix*) [13–16]. A wide range of host tree species are conducive to the invasion and spread of wood wasps, and they increase the difficulty of its prevention and control, which has led to its continuous expansion worldwide for many years. In the last 100 years, with increased human activity and international trade, *S. noctilio* has invaded Oceania (New Zealand and Australia), South America (Uruguay, Argentina, Brazil, and Chile), North America (Canada and the United States), and Asia (China) [17–23], thus becoming a global invasive pest. Sun [24] predicted that every continent except Antarctica has areas suitable for *S. noctilio*. As a result, effective control of *S. noctilio* requires extreme vigilance, because this invasive pest can rapidly multiply and spread in new areas lacking competing species and natural enemies, causing great damage to host plants and serious economic losses [25]. Until now, *S. noctilio* in China has been distributed mainly in the northeast and parts of Inner Mongolia, which is consistent with the highly suitable area predicted by the CLIMEX model [26]. In countries where invasive *S. noctilio* colonization has occurred, parasitic wasps and nematodes have achieved good control effects [27]. However, the implementation of biological control is greatly affected by environmental factors. To achieve real-time monitoring of low-density populations, new methods of controlling *S. noctilio* from the perspective of chemical ecology through attractants are necessary.

Generally, odor molecules with potential utility as wood wasp attractants are divided into three categories: host plant volatiles, symbiotic fungal volatiles, and sex pheromones. One or more odor molecules are reasonably formulated into attractants to regulate the interactions of wood wasps on the environment and achieve the purpose of monitoring or prevention and control. Plant-derived attractants have long been used in forested environments for prevention and control of wood wasps, but their effects are relatively limited. Liu [28] used lure cores with four host plant volatile components to conduct forest experiments in Heilongjiang Province for four consecutive years, but this method was not found to be suitable for monitoring wood wasp populations. Although the use of bait wood to prevent and control wood wasps was in line with their habit of damaging weak host plants, and although it was also suitable for monitoring insect populations with low density, this method is costly and difficult to apply. After discovering the gap in trapping work, there is an urgent need to identify new attractants with improved specificity and optimize the attractant formulation [29]. Whether there is a new perspective that can effectively regulate the interactions between male adults and odor molecules and achieve effective prevention and control effects needs to be studied.

Pheromone control is becoming an important measure in comprehensive pest management due to its high efficiency, nontoxicity, and strong specificity. However, according to existing work [30,31], the use of *S. noctilio* pheromones alone was found to be an unsatisfactory method of forest trapping; thus, pheromones and plant volatiles were often used in combination. Previous studies have mostly focused on the development of male pheromone attractants. For example, Sarvary et al. [32] developed a lure consisting of three male pheromones ((*Z*)-3-decen-1-ol, (*Z*)-4-decen-1-ol, and (*E*,*E*)-2, 4-decadienal), which were found to be effective attractant ingredients in an indoor attractant activity study. However, a study of female pheromones derived from cuticular washes identified the pheromones (*Z*)-7-heptacosene, (*Z*)-7-nonacosene, and (*Z*)-9-nonacosene, and these pheromones were found to be active in laboratory assay experiments and posited to be short-distance contact pheromones, which meant that the actual effective distance must be evaluated in field experiments [33]. In addition, the female adult carries and spreads the symbiotic bacterium *Amylostereum areolatum* [34], which is injected into the tree trunk with a phytotoxin when eggs are laid, causing the host to become debilitated and eventually die. It can be seen that the symbiotic fungus is involved in a crucial part of the reproductive oviposition process of the wood wasps, and its volatiles also perhaps play an important role in the interactions between *S. noctilio* and the environment. Therefore, we combined knowledge from previous experiments and the established habits of wood wasps to develop a new attractant based on female sex pheromones and symbiotic fungal volatiles. In previous studies using antennae transcriptome data and qPCR analysis of tissue expression to identify olfactory-related genes, a total of 16 *SnocOBPs* with different expression levels were obtained. SnocOBP7 was chosen for this study because of its gender-specific expression pattern (it is expressed only in male genitalia) [35], which allowed us to identify gender-specific attractants through reverse chemical ecology and clarify its biological interaction mechanism.

Is SnocOBP7 related to the recognition and interaction of odor molecules? How does it play a regulatory role in protein–ligand interactions? Protein–ligand interactions (PLI) are important processes in organisms. In pest control, they are of great significance for the development and design of attractants and understanding the molecular mechanism of interactions, aiming to clarify the molecular interaction that occurs in the specific binding region. On the basis of the specific expression of SnocOBP7 in the olfactory system of male wood wasps, this study explored the binding interaction relationship between SnocOBP7 and four types of odor molecules: male aggregation pheromones, female sex pheromones, host plant volatiles, and symbiotic fungal volatiles. In order to dynamically characterize the binding between SnocOBP7 and different chemical volatiles and identify the key residues involved in the binding process, we performed 50 ns MD simulations of the ligand-connected complexes. After obtaining stable representative conformations and

identifying the ligands with the best binding to SnocOBP7 [36], we performed binding free energy calculations, pairwise per-residue free energy decomposition, and computational alanine scanning (CAS) to determine their key amino-acid residues. The prediction of key protein binding sites can guide the structural analysis and function prediction of protein complexes, as well as design molecules that can regulate biological functions at the system level [37]. The results of this study demonstrate the presence of functional OBPs with gender-specific expression in *S. noctilio* and provide a foundation for the development of improved methods for prevention and control through studying its protein–ligand interaction mechanism.

## 2. Materials and Methods

### 2.1. RNA Extraction and cDNA Synthesis

Adult samples of *S. noctilio* were collected from five pieces of damaged wood from Xindian Forest Station in Durbert Mongolian Autonomous County, Daqing City, Heilongjiang Province. After the adults emerged and flew for the first time, the external genitalia of the male adults were collected and dissected, immediately frozen in liquid nitrogen, and stored at −80 °C for later use.

RNA was extracted from the frozen male genitalia using the RNeasy Mini Kit (QIAGEN, Hilden, Germany). The quality and concentration of RNA were analyzed using an ultramicro-spectrophotometer (Nanodrop 8000, Thermo, Waltham, MA, USA). First-strand complementary DNA (cDNA) was synthesized using the Prime Script™ RT Reagent Kit with gDNA Eraser Kit (Takara, Dalian, China) following the manufacturer's instructions. The cDNA was subjected to PCR amplification, and all PCR products were stored at −20 °C.

### 2.2. Sequence Analysis of SnocOBP7

The sequence of SnocOBP7 was derived from our previous transcriptome analysis results. Open reading frames (ORFs) and associated amino-acid sequences were determined using the ORF Finder Tool (https://www.ncbi.nlm.nih.gov/orffinder/; accessed on 16 November 2020). The molecular weight, theoretical isoelectric point (pI), and hydrophobicity were calculated using ExPASy (https://web.expasy.org/protparam/; accessed on 3 December 2020 and https://web.expasy.org/protscale/; accessed on 3 December 2020). The SignalP 4.1 program (http://www.cbs.dtu.dk/services/SignalP/; accessed on 15 November 2020) was applied to predict the mature protein sequence and signal peptides, and Loctree3 (https://rostlab.org/services/loctree3/; accessed on 5 December 2020) was used to assess subcellular localization. The sequence data of SnocOBP7 were analyzed using the NCBI BLAST server (http://www.ncbi.nlm.nih.gov/BLAST; accessed on 9 January 2021), and Weblogo3 (http://weblogo.threeplusone.com/create.cgi; accessed on 14 January 2021) was used to visualized multiple alignment results by ClustalX 2.0 [38].

### 2.3. Gene Cloning and Verification of the SnocOBP7 Sequence

The validity of the putative OBPs was confirmed by cloning and sequencing ORFs using corresponding primer pairs. On the basis of the sequence of SnocOBP7, primers were designed using Primer3plus (http://www.primer3plus.com/cgi-bin/dev/primer3plus.cgi; accessed on 10 January 2021) to clone the entire ORF of SnocOBP7 (Table 1).

**Table 1.** Primers used for cloning of SnocOBP7.

| Name | Sequence |
|:---:|:---:|
| SnocOBP7F (5′ to 3′) | GGCGGACATTAGAAGAGACTGT |
| SnocOBP7R (3′ to 5′) | TGCTTCAAGATCTCGGGCTG |

ExTaq DNA polymerase (TransGen, Beijing, China) was used to amplify individual sequences under the following conditions: initial denaturation at 95 °C for 5 min, 94 °C for 30 s, 55 °C for 30 s, then 35 cycles at 72 °C for 30 s, and a final extension at 72 °C for 10 min. The PCR

products were gel-purified using an AxyPrep DNA Gel Extraction Kit (Axygen, Union City, CA, USA) and then subcloned into the pEasy®-T3 vector (TransGen, Beijing, China). Finally, the product was sequenced to identify whether the target fragment was inserted correctly.

### 2.4. Homology Modeling and Molecular Docking

The program SWISS-MODEL (http://swissmodel.expasy.org/; accessed on 13 November 2020) was used to search for the best suitable model template for 3D structure construction [39]. In the results, *Locusta migratoria* OBP1 (PDB ID: 4PTI) was selected for homology modeling because it had the highest identity (32.69%) with SnocOBP7. From 20 models built by Modeller 9.25 [40–43], the best initial model was selected according to the lowest discrete optimized protein energy (DOPE) score and refined by UCSF Chimera1.14 [44,45]. The residue compatibilities and stereochemical rationalities of the model were examined using the online program SAVEs v6.0 (https://saves.mbi.ucla.edu/, accessed on 16 November 2020). Molecular docking was carried out using the AutoDock Tools 1.5.6 package [46]. The required ligand files were downloaded from Pubchem (https://pubchem.ncbi.nlm.nih.gov/; accessed on 25 November 2020) and were divided into three categories: host plant volatiles, male and female pheromones, and symbiotic fungal volatiles. The size of the Autogrid box was determined by Protein plus DoGSiteScore (https://protein.plus/; accessed on 25 November 2020). The optimal conformation of the SnocOBP7–ligand complex was determined by the lowest estimated free energy of binding and subjected to visual analysis using PyMol2.3.0 [47].

### 2.5. Molecular Dynamics (MD) Simulations of SnocOBP7 and 11 Ligands

After obtaining protein conformations as described above, all MD simulations of SnocOBP7–ligand complexes were carried out using the GROMACS2019.6 software package [48] with the AMBER-ff99sb-ildn force field [49] for protein SnocOBP7 and the AmberTools 18 GAFF force field [50], which was optimized with the ACPYPE script [51] for ligands. Ligand information is listed in Table 2 and Figure 1. First, $Na^+$ or $Cl^-$ ions were added according to the total charge of the system. For example, the SnocOBP7–(Z)-7-heptacosene complex contained 6082 TIP3P water molecules and the total charge was +8; hence, eight $Cl^-$ ions were added for charge neutralization. System energy minimization was achieved by a conjugated gradient (CG) method, after which NVT and NPT (N means the number of particles, P means pressure, T means temperature, and any letter in the name means the value is constant) ensembles for position-restricted MD simulations were performed to relax water solvents. MD simulation was performed for 50 ns with a time step of 2 fs with constant temperature (298.15K) and pressure (1 bar) coupling using a V-rescale thermostat and a Parrinello–Rahman barostat, respectively. The particle mesh Ewald (PME) algorithm was used to calculate the long-range electrostatic interactions, while the LINCS algorithm was applied to constrain all covalent bonds with hydrogen atoms. The cutoff radii of both Coulomb and van der Waals interactions were set to 10 Å. The coordinate information of all systems was recorded every 2 ps, and 25,000 configurations were dumped into the corresponding trajectories for an analysis of the correlations between the conformations. The equilibrium of SnocOBP7 with 11 ligands was analyzed according to the degree of curve fluctuation of the root-mean-square deviation (RMSD). The gmx_cluster module was used for cluster analysis to distinguish different structures and define the representative dominant conformations [52].

### 2.6. Binding Free Energy Calculation and Per-Residue Free Energy Decomposition

The molecular mechanics/Poisson–Boltzmann surface area (MM/PBSA) method [53] and g_mmpbsa script [54] were selected to calculate the free binding energy of SnocOBP7 with 11 ligands. The energy contribution of each residue was further decomposed into van der Waals and electrostatic energy, polar solvation free energy, and nonpolar solvation free energy. The binding free energy ($\Delta G_{bind}$) was defined as follows:

$$\Delta G_{bind} = \Delta G_{complex} - \Delta G_{receptor} - \Delta G_{ligand}, \tag{1}$$

$$\Delta G_{bind} = \Delta E_{MM} + \Delta G_{sol} - T\Delta S, \tag{2}$$

$$\Delta E_{MM} = \Delta G_{internal} + \Delta G_{ele} + \Delta G_{vdw}, \tag{3}$$

$$\Delta G_{sol} = \Delta G_{PB} + \Delta G_{SA}. \tag{4}$$

The binding free energy ($\Delta G_{bind}$) was equal to the difference in energy of the complex ($\Delta G_{complex}$), receptor protein($\Delta G_{receptor}$), and the ligand ($\Delta G_{ligand}$); the binding free energy ($\Delta G_{bind}$) was evaluated by calculating the sum of the molecular mechanics energy ($\Delta E_{MM}$), the solvation free energy ($\Delta G_{sol}$), and the conformational entropy effect on binding ($-T\Delta S$) in the gas phase; $\Delta E_{MM}$ was the sum of the potential energy of the internal bond energy ($\Delta G_{internal}$), electrostatic energy ($\Delta G_{ele}$), and van der Waals energy ($\Delta G_{vdw}$); $\Delta G_{PB}$ and $\Delta G_{SA}$ were the solvation free energies of polar and nonpolar solvents, respectively. In addition, $-T\Delta S$ was neglected because of high computational cost [55], and the residues of proteins with free energy contributions greater than $-1.00$ kcal·mol$^{-1}$ can be considered as key contributors to binding affinity [56].

**Table 2.** Main properties of 11 ligands.

| ID | Ligand Name | Molecular Formula | PubChem ID | Molecular Weight (g/mol) | Complex |
|---|---|---|---|---|---|
| | **Female pheromones** | | | | |
| P1 [A] | (Z)-7-heptacosene | $C_{27}H_{54}$ | 56936088 | 378.7 | S1 [C] |
| P2 | (Z)-7-nonacosene | $C_{29}H_{58}$ | 56936089 | 406.8 | S2 |
| | **Male pheromones** | | | | |
| P3 | (Z)-4-decen-1-ol | $C_{10}H_{20}O$ | 5362798 | 156.26 | S3 |
| P4 | (E,E)-2,4-decadienal | $C_{10}H_{16}O$ | 5283349 | 152.23 | S4 |
| P5 | (Z)-3-decen-1-ol | $C_{10}H_{20}O$ | 5352846 | 156.26 | S5 |
| | **Host plant volatiles** | | | | |
| V6 [B] | Camphene | $C_{10}H_{16}$ | 6616 | 136.23 | S6 |
| V7 | Eucalyptol | $C_{10}H_{18}O$ | 2758 | 154.25 | S7 |
| V8 | (−)-Limonene | $C_{10}H_{16}$ | 439250 | 136.23 | S8 |
| | **Fungal volatiles** | | | | |
| V9 | 1-(3-ethylphenyl)ethanone | $C_{10}H_{12}O$ | 31493 | 148.2 | S9 |
| V10 | (−)-Globulol | $C_{15}H_{26}O$ | 12304985 | 222.37 | S10 |
| V11 | Linalool | $C_{10}H_{18}O$ | 6549 | 154.25 | S11 |

[A] P1 refers to Sex Pheromone 1. [B] V6 refers to Volatile 6. [C] S1 refers to the corresponding simulation of SnocOBP7 and P1. Other ligands or systems are named in the same way.

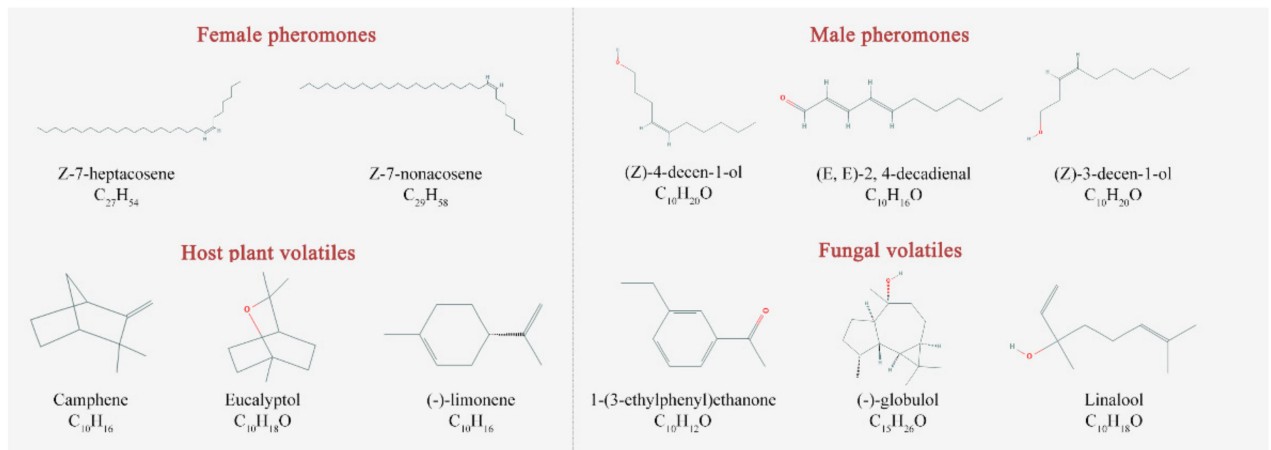

**Figure 1.** The 2D structures of 11 ligands. Refer to Table 2 for details.

### 2.7. Computational Alanine Scanning (CAS)

Computational alanine scanning (CAS) is an energy-based method of identifying key amino-acid residues involved in protein–ligand binding [57], and it can be used to illustrate the contributions of key contributors to binding based on the results of MM/PBSA. All alanine mutation calculations were performed using Discovery Studio Client 2019 (DS,

Accelrys Inc., USA) according to the publisher's protocol. The calculation principle was as follows:

$$\Delta\Delta G_{mut} = \Delta G_{bind} \text{ (mutant)} - \Delta G_{bind} \text{ (wild-type)}. \tag{5}$$

In this formula, $\Delta\Delta G_{mut}$ is the binding free energy difference between the wild-type and mutants, $\Delta G_{bind}$ (mutant) is the binding free energy of the mutants, and $\Delta G_{bind}$ (wild-type) is the energy of the wild-type. From the value of $\Delta\Delta G_{mut}$, the effect of the mutant residue on the binding affinity can be determined. If $\Delta\Delta G_{mut}$ is between −0.5 and 0.5, the mutant residue has no significant effect on the affinity; if it is greater than 0.5, the mutant residue weakens the interaction and reduces affinity; if it is lower than −0.5, the mutant residue enhances the binding affinity for the protein–ligand interaction. Finally, we selected residues with $\Delta\Delta G_{mut}$ greater than 0.5 and analyzed their effects on the protein–ligand system [58].

## 3. Results

### 3.1. Sequence Analysis and Homology Modeling

After obtaining the complete sequence of SnocOBP7, ORFs were identified and BLASTP analysis was performed. Five OBPs from other Hymenoptera species were selected for multiple sequence alignment with SnocOBP7 (specific information is shown in Figure 2). The results showed that SnocOBP7 contains 131 amino acids with six conserved cysteine residues. Amino acids 1–22 at the N-terminus are the signal peptide region (Figures 2a and 3). According to the results of the bioinformatics analysis, SnocOBP7 is a small-molecule hydrophobin with a molecular weight of 15.29 ku, which performs secretory functions outside the cell. In summary, SnocOBP7 conforms to the basic characteristics of OBPs. In order to find the most suitable template to create the 3D structure of SnocOBP7, we first identified *Locusta migratoria* OBP1 (PDB ID: 4PTI) (Figure 2b) as the currently available template with the highest homology (identity = 32.69%), and it was used as the initial template for optimization. Finally, we obtained an optimized model of SnocOBP7 (Figure 2c).

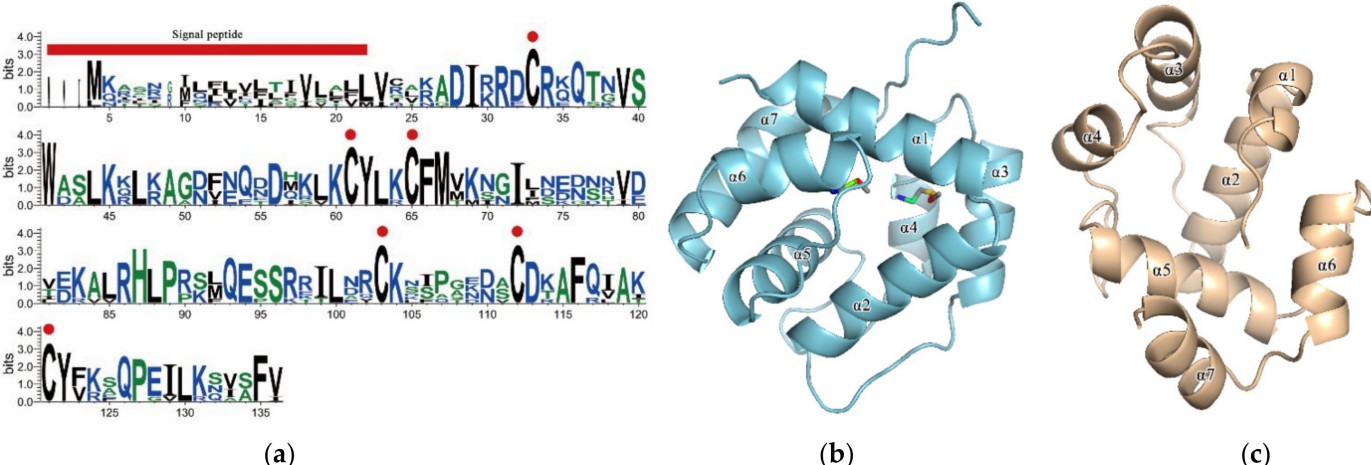

(a)    (b)    (c)

**Figure 2.** Sequence analysis and structural modeling of SnocOBP7. (**a**) SnocOBP7 sequence multiple alignment results of *S. noctilio* and other Hymenoptera. The amino acids which only exist at one site are highly conserved, and the height of the symbol within the stack indicates the relative frequency of each amino group at that position. Red dots indicate conserved cysteines. The insect species and GenBank accession numbers used for the comparison were as follows: *Sirex nitobei*: OBP7 (QHN69064.1); *Cephus cinctus*: OBP10 (ARN17866.1); *Harpegnathos saltator*: GOBP83a (XP_019697001.2); *Camponotus floridanus*: GOBP83a (XP_011254774.2); *Solenopsis invicta*: GOBP83a (XP_011173007.1); *Aulacocentrum confusum*: OBP8 (QNL14934.1). Each logo consisted of stacks of symbols (one stack for each position in the sequence). (**b**,**c**) Structural modeling of SnocOBP7: (**b**) *Locusta migratoria* OBP1 (PDB ID:4PTI); (**c**) optimized SnocOBP7 model.

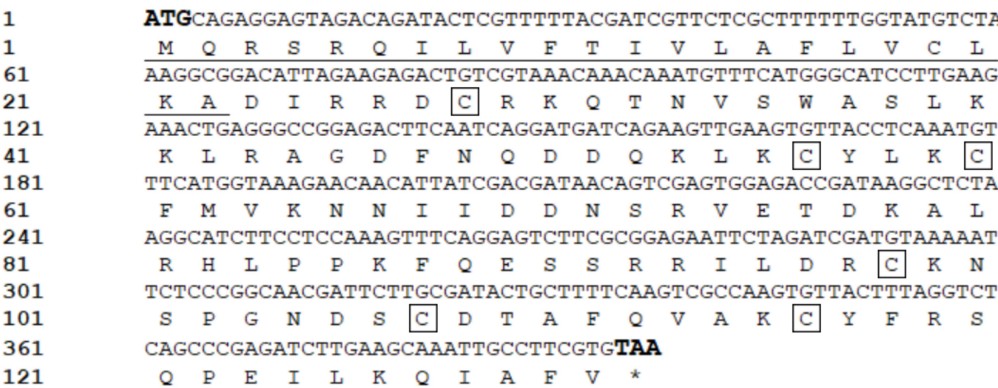

**Figure 3.** Nucleotide sequence and deduced amino-acid sequence of SnocOBP7. The start and stop codons are shown in bold type, the sequence of the signal peptide is underlined, and conserved cysteine residues are bound by boxes.

Before molecular docking, the quality of the model must be assessed to ensure its usability. As shown in the Ramachandran plot (Figure S1a), 92.91% of the amino-acid residues of the SnocOBP7 model were located in the most favored regions (A, B, L, the red region in the figure), and all non-Gly residues were located in the allowed regions. In addition, the overall quality factor of ERRAT was 98.0198, and 88.99% of the residues were found to meet the Verify_3D standard, which strongly suggested that the model was of high quality (Figure S1b,c). By observing the 3D protein structure of SnocOBP7, the model was found to possess seven α-helices and a hydrophobic cavity. Classical OBPs generally have several common structural characteristics, including six or seven α-helices, three disulfide bonds, and a hydrophobic binding cavity [59]. In addition, in classical OBPs, the α7 helix at the C-terminus forms the wall of the hydrophobic binding cavity.

### 3.2. Stability of SnocOBP7–Ligand Complexes in MD Simulation

The putative binding pocket of SnocOBP7 was predicted in the 3D model (shown in the yellow, green, and purple grid area) (Figure S2). The yellow area in the hydrophobic cavity was identified as the docking area where the SnocOBP7–ligand complexes were generated. According to the time-evolution root-mean-square deviation (RMSD) curves of all 11 systems (Figure 4), most of the SnocOBP7–ligand complex systems could eliminate spatial conflicts and reach equilibrium at 20 ns, and all systems converged within 50 ns, indicating that 50 ns MD simulations have practical significance and can be used as a basis for exploring the global conformation of SnocOBP7. The average RMSD value of the 11 systems ranged from 2.7 Å to 3.5 Å. The RMSD curve stabilized after 20 ns; the maximum value of the standard deviation of the RMSD was 0.23 Å, and the minimum value was only 0.09 Å. The root-mean-square fluctuation value (RMSF) reflects the local motion characteristics and degree of freedom of the protein secondary structure elements when combined with the ligand (Figure S3). The RMSF curve showed that the largest fluctuation was in α4 (amino acids between 54 and 60) and α5 (amino acids between 66 and 76), and there was a common peak in the 60–65 residue region (loop and part of α5); the fluctuations in the N-terminus and C-terminus were also large. The maximum RMSF values of S5, S6, S9, and S11 were all located at the 71st amino acid (arginine).

Cluster analysis of the complexes was performed, and the complex with the highest rate of occurrence was selected as the representative conformation for free binding energy decomposition [60,61]. The free binding energy ($\Delta G_{bind}$) of the SnocOBP7 complexes was decomposed into the van der Waals energy ($\Delta G_{vdw}$), the electrostatic energy ($\Delta G_{ele}$), the polar solvation energy ($\Delta G_{PB}$), and the apolar solvation energy ($\Delta G_{SA}$) (Table 3). The $\Delta G_{bind}$ values of S1 ($-55.656 \pm 0.351$ kcal·mol$^{-1}$) and S2 ($-56.783 \pm 0.260$ kcal·mol$^{-1}$) were significantly greater than those of the other complexes, and the $\Delta G_{bind}$ of S10 was also relatively high ($-31.057 \pm 0.154$ kcal·mol$^{-1}$). Among the four types of calculated energy,

$\Delta G_{vdw}$ had the highest value (as high as $-62.877 \pm 0.220$ kcal·mol$^{-1}$ in S2), indicating that it was the main force promoting the formation of SnocOBP7 complexes [62]. $\Delta G_{ele}$ and $\Delta G_{SA}$ were weak, but they still played a positive role in SnocOBP7 complex formation; in general, the effect of $\Delta G_{SA}$ was slightly stronger than that of $\Delta G_{ele}$. $\Delta G_{PB}$ was greater than 0, which showed that it had a negative effect and inhibited the formation of SnocOBP7 complexes. When the side-chains of nonpolar amino acid residues form the three-dimensional structure of the protein, they twist and fold together to form the nonpolar region of the active site, which is known as the hydrophobic pocket. Polar residues were generally located in the opening or bottom of the pocket due to their hydrophilicity, which was why $\Delta G_{PB}$ inhibited binding between SnocOBP7 and ligands.

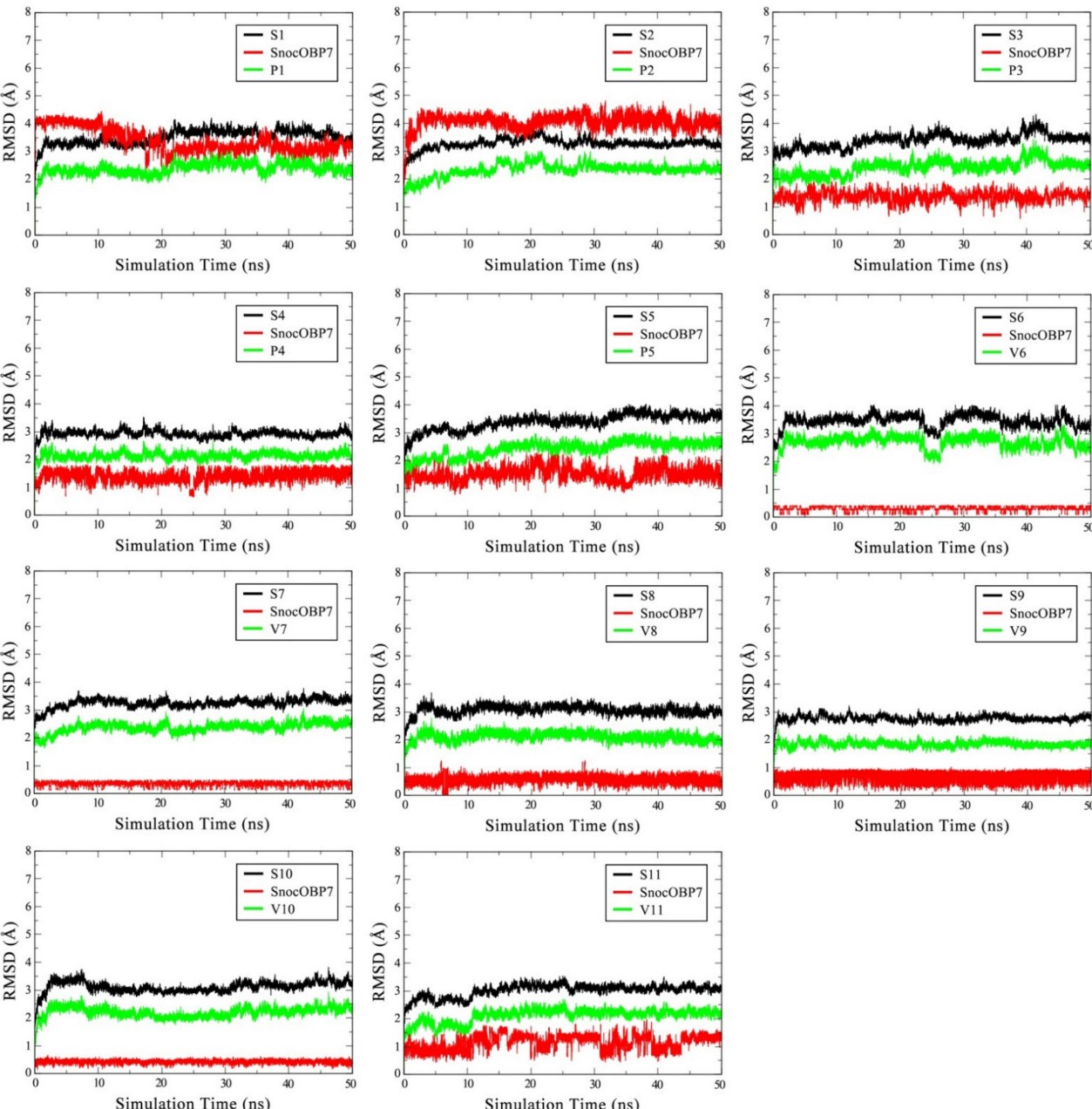

**Figure 4.** Molecular dynamics (MD) simulations (50 ns) of the SnocOBP7–ligand complexes. The time-evolution RMSD curves of 11 systems are shown. The curves of the complex are shown in black, the curves of protein are shown in red, and the curves of the ligands are shown in green. Refer to Table 1 for specific information regarding the 11 systems and ligands shown here.

**Table 3.** The binding free energy of SnocOBP7–ligand complexes.

| Complex | Cluster (ns) | Van der Waals Energy ($\Delta G_{vdw}$) | | | Electrostatic Energy ($\Delta G_{ele}$) | | | Polar Solvation Energy ($\Delta G_{PB}$) | | | SASA Energy ($\Delta G_{SA}$) | | | Binding Energy ($\Delta G_{bind}$) | |
|---|---|---|---|---|---|---|---|---|---|---|---|---|---|---|---|
| S1 | 21–50 | −59.900 | ± | 0.341 | −0.300 | ± | 0.018 | 10.975 | ± | 0.188 | −6.432 | ± | 0.034 | −55.656 ± | 0.351 |
| S2 | 23–50 | −62.877 | ± | 0.220 | −0.385 | ± | 0.027 | 13.013 | ± | 0.196 | −6.533 | ± | 0.018 | −56.783 ± | 0.260 |
| S3 | 25–50 | −26.014 | ± | 0.162 | −1.547 | ± | 0.097 | 9.895 | ± | 0.146 | −3.071 | ± | 0.010 | −20.740 ± | 0.150 |
| S4 | 25–50 | −30.342 | ± | 0.130 | −2.300 | ± | 0.068 | 15.082 | ± | 0.081 | −3.121 | ± | 0.010 | −20.668 ± | 0.140 |
| S5 | 33–50 | −29.697 | ± | 0.123 | −5.350 | ± | 0.140 | 14.269 | ± | 0.141 | −3.372 | ± | 0.010 | −24.143 ± | 0.184 |
| S6 | 27–48 | −24.627 | ± | 0.099 | −0.108 | ± | 0.012 | 4.518 | ± | 0.051 | −2.540 | ± | 0.008 | −22.755 ± | 0.111 |
| S7 | 25–40 | −26.805 | ± | 0.105 | −0.339 | ± | 0.031 | 5.513 | ± | 0.045 | −2.721 | ± | 0.009 | −24.347 ± | 0.111 |
| S8 | 25–50 | −24.903 | ± | 0.092 | −0.095 | ± | 0.016 | 7.564 | ± | 0.057 | −2.795 | ± | 0.008 | −20.228 ± | 0.107 |
| S9 | 25–49 | −25.514 | ± | 0.104 | −4.347 | ± | 0.094 | 13.277 | ± | 0.065 | −2.776 | ± | 0.009 | −19.366 ± | 0.112 |
| S10 | 21–50 | −34.180 | ± | 0.130 | −3.947 | ± | 0.127 | 10.632 | ± | 0.087 | −3.562 | ± | 0.009 | −31.057 ± | 0.154 |
| S11 | 25–40 | −26.293 | ± | 0.132 | −3.541 | ± | 0.115 | 12.040 | ± | 0.114 | −3.072 | ± | 0.008 | −20.867 ± | 0.163 |

All values are given in kcal·mol$^{-1}$, with corresponding standard errors of the mean in parentheses.

### 3.3. Binding Mode Analysis of SnocOBP7-Ligand Complexes

According to the results described above, we selected the three systems with the lowest free binding energy (S1, S2, and S10) for subsequent analysis. Using cluster analysis, several clusters with different frequencies that appeared during the process of protein binding with ligands were obtained. The most frequently appearing cluster was regarded as the typical conformation of the system. S1, S2, and S10 generated nine, eight, and five clusters, respectively, and the appearance rates of the most common clusters were 38.21%, 39.53%, and 66.11%, respectively (Table S1). The most commonly appearing cluster (cluster 1) of each system was selected for binding mode analysis and combined with the key amino-acid residues predicted by MM-PBSA for visual display (Table 4, Figure 5).

**Table 4.** Free binding energy decomposition for important residues contributing to SnocOBP7–ligand binding.

| Complex | Residue | $\Delta G_{MM}$ | $\Delta G_{PB}$ | $\Delta G_{SA}$ | $\Delta G_{bind}$ |
|---|---|---|---|---|---|
| **S1** (21–50 ns) | Leu36 | −1.673 | 0.534 | −0.121 | −1.260 |
| | Phe39 | −1.427 | 0.374 | −0.106 | −1.161 |
| | Met40 | −2.041 | 0.302 | −0.159 | −1.898 |
| | Ile45 | −1.173 | 0.037 | −0.108 | −1.244 |
| | Leu61 | −1.183 | 0.058 | −0.182 | −1.309 |
| | Ile102 | −1.816 | 0.344 | −0.156 | −1.626 |
| | Ile106 | −3.076 | 0.137 | −0.435 | −3.374 |
| | Phe108 | −1.561 | 0.385 | −0.098 | −1.275 |
| **S2** (23–50 ns) | Leu36 | −1.535 | 0.313 | −0.160 | −1.382 |
| | Phe39 | −1.832 | 0.893 | −0.153 | −1.092 |
| | Leu61 | −0.949 | −0.068 | −0.064 | −1.081 |
| | Leu103 | −2.020 | 1.116 | −0.226 | −1.131 |
| | Phe108 | −3.534 | 2.188 | −0.395 | −1.744 |
| **S10** (21–50 ns) | Met40 | −1.690 | 0.612 | −0.129 | −1.208 |
| | Ile45 | −1.455 | 0.078 | −0.094 | −1.470 |
| | Leu61 | −1.121 | −0.007 | −0.115 | −1.243 |
| | Ile106 | −1.218 | 0.020 | −0.173 | −1.371 |

All values are given in kcal·mol$^{-1}$.

As shown in Figure 5, the hydrophobic cavity of S1 was composed of eight amino-acid residues (Leu36, Phe39, Met40, Ile45, Leu61, Ile102, Ile106, and Phe108), whereas that of S2 was composed of five amino acid residues (Leu36, Phe39, Leu61, Leu103, and Phe108), and that of S10 was composed of four amino acid residues (Met40, Ile45, Leu61, and Ile106). Leu, Phe, Ile, and Met were all nonpolar amino acids. These hydrophobic residues were located inside the protein and formed a hydrophobic core facing all directions, maintaining the tight structure of SnocOBP7. The ligands were firmly anchored in the hydrophobic binding cavity; in other words, SnocOBP7 and the ligands could bind stably in it. The side chains of nonpolar amino-acid residues were the structural basis for the formation of

hydrophobic interactions, and the hydrophobic force was the main force for ligand binding. Therefore, follow-up studies on these nonpolar residues were conducted.

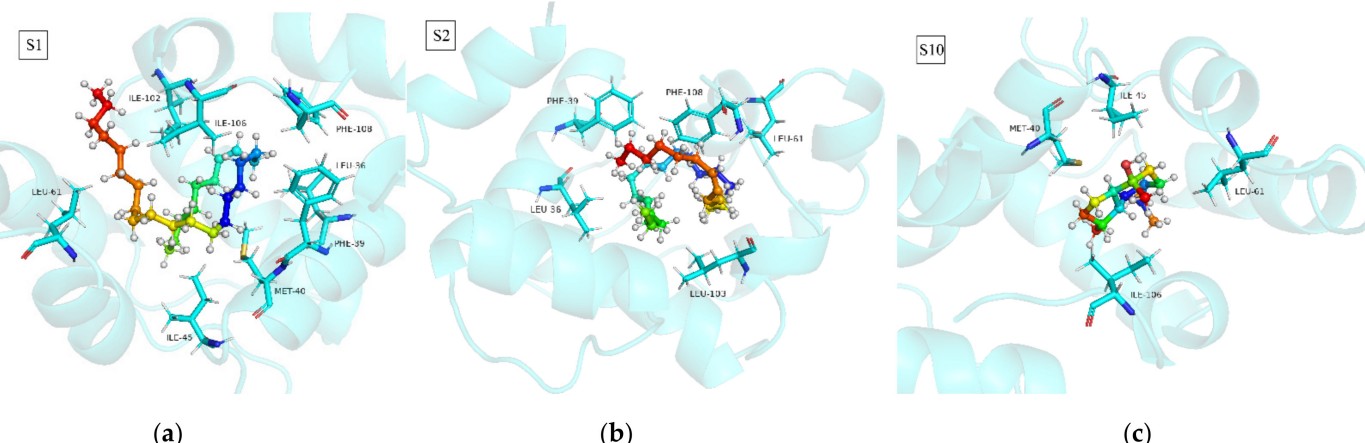

**Figure 5.** Representative conformation (Cluster 1) of the SnocOBP7–ligand complexes (S1, (**a**); S2, (**b**); S10, (**c**)) produced on the basis of the MD simulation trajectories. Representative hydrophobic residues on the binding interface are marked. The ligands are displayed in the form of a sphere–stick model.

By observing the local motion characteristics of amino-acid residues, it was found that the RMSF values of the abovementioned binding residues were generally low in the 50 ns MD simulation. The minimum RMSF value of S1 was only 0.356 Å (Leu36), whereas that of S2 was 0.361 Å (Leu36), and that of S10 was 0.424 Å (Met40); each of these values was calculated for the common depression formed by residues 36 to 40. The other key residues were also located in the concave area of the RMSF curve shown in Figure 6. The RMSF values at other sites, such as the N-terminus, the C-terminus, and a small peak near Lys64, were higher and unstable. In summary, these key amino-acid residues could interact with small ligand molecules and form stable complexes.

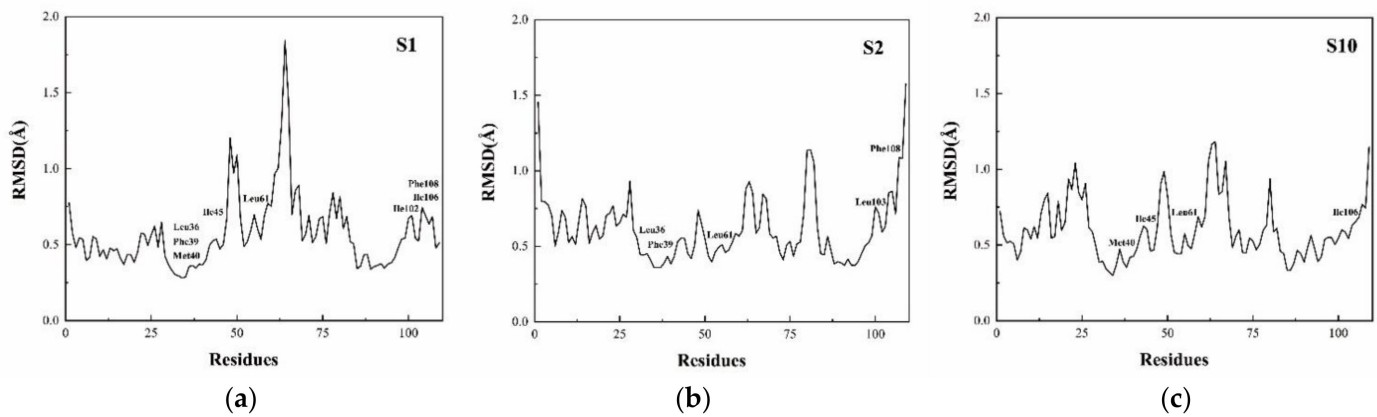

**Figure 6.** The root-mean-square fluctuation (RMSF) curve of SnocOBP7–ligand complexes (S1, (**a**); S2, (**b**); S10, (**c**)).

### 3.4. Per-Residue Free Energy Decomposition

The MM-PBSA, which has been widely used to evaluate the binding free energy of complexes, was used to calculate the contribution of each amino-acid residue of the SnocOBP7–ligand complex to its binding free energy. As shown in Figure 7 and Table 4, there were eight (Leu36, Phe39, Met40, Ile45, Leu61, and Ile102), five (Leu36, Phe39, Leu61, Leu103, and Phe108), and four (Met40, Ile45, Leu61, and Ile106) key amino-acid residues that contributed more than $-1$ kcal·mol$^{-1}$ to the binding free energy for S1,

S2, and S10, respectively. Therefore, these key amino acids were selected for energy decomposition analysis.

**Figure 7.** Per-residue contribution to the binding free energy of SnocOBP7–ligand complexes (S1, (**a**); S2, (**b**); S10, (**c**)) calculated from the equilibrated conformations. The residues contributing more than $-1.00$ kcal/mol to the binding free energy are marked by the red line.

The highest value of $\Delta G_{bind}$ was $-3.374$ kcal·mol$^{-1}$ and the lowest was $-1.08$ kcal·mol$^{-1}$. As shown in Table 4, one amino-acid residue (Ile106 of S1) had a $\Delta G_{bind}$ value that exceeded $-3$ kcal·mol$^{-1}$ because of its high $\Delta E_{MM}$ ($-3.076$ kcal·mol$^{-1}$). The $\Delta E_{MM}$ of Phe108 in S2 also exceeded $-3$ kcal·mol$^{-1}$; however, due to the strong inhibitory $\Delta G_{PB}$ (2.188 kcal·mol$^{-1}$), the final $\Delta G_{bind}$ was only $-1.744$ kcal·mol$^{-1}$. $\Delta G_{PB}$ was mostly positive, which was unfavorable for binding, while $\Delta G_{SA}$ was negative and had a positive, but weak effect on binding; $\Delta E_{MM}$ was the main driving force for binding.

### 3.5. Computational Alanine Scanning (CAS)

Computational alanine scanning is an effective tool for evaluating the free energy change caused by the substitution of amino acids with alanine (Ala) residues [63]. Therefore, we used the key residues of S1, S2, and S10 obtained above as the target for CAS to determine the potential for these residues to participate in binding ligands. As shown in Table 5, replacement of five residues of S1 (Leu36, Met40, Leu61, Leu73, and Phe108five 5 residues of S2 (Leu36, Phe39, Leu61, Leu103, and Phe108), and two residues of S10 (Met40 and Leu61) with Ala resulted in a mutation energy ($\Delta\Delta G_{mut}$) change greater than 0.5 kcal·mol$^{-1}$. These residues were also identified by the per-residue free energy decomposition analysis, demonstrating that the results were consistent and reliable. The $\Delta\Delta G_{mut}$ values of other residues from the per-residue free energy decomposition analysis results were less than 0.5 kcal·mol$^{-1}$, indicating that replacement of these residues with alanine had little effect on the binding affinity of the SnocOBP7 protein with ligands. According to the DS assessment criteria [64,65], the residues identified via CAS may have important effects in stabilizing SnocOBP7–ligand complexes, and amino-acid changes at these positions would be expected to change the active conformation of SnocOBP7, ultimately changing the ability of the protein to bind ligands.

**Table 5.** Computational alanine scanning (CAS) identified important residues contributing to SnocOBP7–ligand binding.

| Complex | Mutation | $\Delta\Delta G_{mut}$ |
|---------|----------|------------------------|
| S1 | Leu36>Ala | 0.77 |
| | Phe39>Ala | 0.46 |
| | Met40>Ala | 1.6 |
| | Ile45>Ala | 0.32 |
| | Leu61>Ala | 1.26 |
| | Ile102>Ala | 0.24 |
| | Ile106>Ala | 0.38 |
| | Phe108>Ala | 0.85 |
| S2 | Leu36>Ala | 0.88 |
| | Phe39>Ala | 1.55 |
| | Leu61>Ala | 1.55 |
| | Leu103>Ala | 1.75 |
| | Phe108>Ala | 2.63 |
| S10 | Met40>Ala | 0.75 |
| | Ile45>Ala | 0.47 |
| | Leu61>Ala | 0.84 |
| | Ile106>Ala | 0.25 |

All values are given in kcal·mol$^{-1}$.

## 4. Discussion

It is well established that differential expression of OBPs is associated with differences in physiological structure and physiological function, and gender-specific OBPs have significant promise in interactions with odor molecules and pest control and monitoring applications. In insects, OBPs are most abundant in the antennae, but specific expression of OBPs also occurs in the abdomen, head, and external genitalia. Several studies have shown that differential expression of OBPs is widespread in Hymenoptera insects. For example, in *Cotesia vestalis*, six OBPs were found to be highly expressed in the antennae of females, while three different OBPs were expressed in the antennae of males [66]. However, the three OBPs of *Sclerodermus alternatusi* were found to be expressed only in the abdomen of female adults; hence, it was speculated that they played a specific role in oviposition behavior [67]. In this study, due to the gender and tissue specificity of SnocOBP7, we speculated that it may play an important role in guiding male adults to complete reproductive mating. Therefore, with the aim of establishing new directions for the development of compounds to control and monitor *S. noctilio*, we used our previous antenna transcriptome data to identify OBP7 as a gender-specific OBP expressed only in the external genitalia of male adults. Subsequently, bioinformatics analysis and molecular dynamics simulations were carried out to reveal the interaction mechanism between SnocOBP7 and various odor molecules.

To explore interaction avenues for improving *S. noctilio* attractants, binding between SnocOBP7 and 11 types of odor molecules (including host plant volatiles, symbiotic fungal volatiles, and sex pheromones) was simulated. Subsequently, energy decomposition analysis of the SnocOBP7–ligand complexes was carried out. Firstly, the screening results showed that (Z)-7-heptacosene and (Z)-7-nonacosene had the highest binding affinity for SnocOBP7 among female wood wasp sex pheromones. The strategies used by *S. noctilio* to locate mates during courtship are not well understood, mainly because mating generally occurs in the high canopy layer and is, thus, difficult to observe. Most studies have indicated that male and female wood wasps are attracted by host plant volatiles [68] or light [69] and, therefore, congregate in the same areas in the upper canopy during mating. After meeting in the upper canopy, male and female wood wasps must locate each other through various signals to increase the possibility of successful mating. Before mating, the male always approaches from behind the female and touches the end of the female's body with his genitalia [70], which suggests that the male wood wasps use their genitalia to recognize females. This is a specific behavioral interaction that odor molecules regulate

insect physiology and then reflect behavior. According to current research progress, female sex pheromones were predicted to be short-distance pheromones. To explore the scope of its effectiveness in the actual application process, it was necessary to carefully adjust the attractant formula and additive dosage in the follow-up work. If the action distance was short, it was compounded with plant volatiles to observe whether it could achieve a close-range attracting effect and enhance the overall effect. Moreover, previous studies have also indicated that female sex pheromones were obtained from female adult body surface extraction. Our findings suggest that (Z)-7-heptacosene and (Z)-7-nonacosene might be used by male *S. noctilio* to locate females; therefore, these female pheromones could guide the development of gender-specific attractants.

Secondly, among the tested symbiotic fungal volatiles, the binding affinity of (−)-globulol far exceeded that of the volatiles, which may be linked to the mating effect of symbiotic bacteria and male adults and an important factor in the regulation of biological interactions. The symbiotic fungus of *S. noctilio* is *A. areolatum*, which has weak pathogenicity but can degrade the lignocellulose of host plant and facilitate larval feeding [71]. (−)-Globulol is notable for its attractiveness to female adult *S. noctilio* [72], but relevant behavioral experiments have not been performed using male adults. Our computer simulations demonstrate that (−)-globulol is capable of binding strongly with male-specific SnocOBP7. Female adults inject eggs and symbiotic fungi into the host plant at the same time when laying eggs. Therefore, it is likely that SnocOBP7, which is expressed only in male genitalia, binds to symbiotic fungal volatiles such as (−)-globulol to signal to adult males that females could be present. This reminds us that, in the follow-up control work, it may be possible to influence the behavioral interaction between male adults and the environment by adjusting the expression of (−)-globulol, so as to achieve the purpose of attracting and trapping. Males may be more inclined to fly to areas with a high density of symbiotic fungal volatiles, because such areas are likely to contain weakened host plants and more mating resources. Symbiotic fungal volatiles are released from the fungal mycangium of females or at oviposition sites on damaged trees [73]. Therefore, the fungal volatile (−)-globulol may also be used in the courtship process of wood wasps as an attractant. *S. noctilio* and its symbiotic fungi have gradually formed a highly evolved and strict obligate dependency mutualism [74]. Because *S. noctilio* locates suitable egg-laying locations through symbiotic fungi, it is possible to mix symbiotic fungal volatiles into attractants formulated with plant-derived volatiles and pheromones to improve their control efficiency. (−)-Globulol, which has a strong binding affinity with males and has been proven to be effective in attracting females, may be an important regulatory odor factor that has been neglected and has not been used in field traps. Therefore, analysis of simulations of SnocOBP7–ligand binding interactions could lead to the development of new strategies to limit the reproduction of *S. noctilio* via intervention at the courtship stage.

Lastly, in our simulation analysis, SnocOBP7 did not bind strongly with the selected male aggregation pheromone, presumably because the antennae of adults generally perceive this type of pheromone. Adult males can cluster in the wild [75], mainly because they produce aggregation pheromones, which cause individuals of the same sex to gather. These male pheromones should play similar role to host volatiles that attract both sexes to gather in the high canopy layer. Different OBPs have different interactions with the environment and have different sensitivity to different odor molecules, which are all related to their expression levels in different tissues.

Different types of free binding energy have different numerical values, which represent different contributions to binding affinity. Among the four energies we analyzed, the value of $\Delta G_{vdw}$ was obviously the highest, indicating that it was the main force driving binding [76]. Indeed, $\Delta G_{vdw}$ was higher than the final $\Delta G_{bind}$ because the negative offset effect of $\Delta G_{PB}$ on the binding could not be ignored. In contrast, although $\Delta G_{ele}$ and $\Delta G_{SA}$ played a role in promoting binding, their contributions were relatively weak. The decomposition of free binding energy also showed that stable SnocOBP7–ligand binding was attributed to the hydrophobic interaction between SnocOBP7 and ligands. OBPs have

a 3D hydrophobic binding cavity, and hydrophobic odor molecules are usually anchored in it to achieve relatively stable binding with proteins. For example, analysis of the binding energy of *A. lepigone* AlepPBP1 showed that the main driving force promoting its binding was the van der Waals force, demonstrating the key role of hydrophobicity [77]. In the CpomPBP2–Dod system of *Cydia pomonella*, in addition to the dominance of the van der Waals force, the electrostatic force also plays a significant role due to the formation of strong salt bridges and H-bonds [78].

Clarifying the binding mechanism of OBP and odor molecules and deciphering their biological interactions with the environment has always been a hot topic in the field of OBP research. The per-residue free energy decomposition showed that each system had several key amino-acid residues, which together formed a hydrophobic binding cavity. For example, the eight amino-acid residues of S1 that formed a hydrophobic binding cavity were Leu36, Phe39, Met40, Ile45, Leu61, Ile102, Ile106, and Phe108. As shown in Figure 5, the ligand was surrounded by these residues, allowing it to be pulled in different directions to achieve balanced binding. Among these amino-acid residues, only the $\Delta G_{bind}$ of Ile106 was greater than $-3$ kcal·mol$^{-1}$, and $\Delta E_{MM}$ was the main contributing force. From the perspective of overlapping key amino-acid positions, Phe108 and Leu36 appeared as key amino acids in S1 and S2. Therefore, it could be inferred that they were the key sites involved in binding of OBP7 with female sex pheromones. Met40 appeared in both S1 and S10, suggesting that it plays a key role in binding to female sex pheromones and symbiotic fungal volatiles; therefore, Met40 could play an important role in regulating the courtship behavior of males. Leu61 was identified in all three systems, suggesting that it could play key roles in mating and host plant identification. More importantly, after obtaining the prediction results of the key amino-acid binding sites, it is possible to induce mutations at these sites from the perspective of the protein SnocOBP7 to achieve the reverse regulation of the biological interactions. Therefore, CAS was performed on the key residues derived from the results described above. If a particular amino-acid residue was replaced by alanine and had a significant effect on the binding energy, then it was determined to play a key role in ligand binding. The key residues, Met40, Leu61, and Ile106, were found to be in the same region, which contained a depression according to the RMSF analysis, which is consistent with our simulation and prediction results. Similarly, in a study of *Anopheles gambiae* AgOBP1, molecular dynamics analysis confirmed that eugenyl acetate was a better insect repellent than DEET and also revealed the main features of the binding site of AgOBP1 [79].

According to reverse chemical ecology, after obtaining the key amino-acid residues and odor ligands for binding, OBPs can be used as a major entry point for the development of new and environmentally friendly attractants and as a powerful tool for interfering with or regulating biological interactions. However, although female pheromones were obtained from the body wall of female adults through chemical leaching in previous studies, actual molecular experiments are needed to determine their potential utility in control and monitoring applications. For example, in a study of the olfactory function of OBP2 in *Apis cerana*, site-directed mutagenesis was used to verify the key amino-acid binding sites predicted by molecular docking [80]. In this study, the results of our computer simulation provide new ideas for explaining the protein-binding interaction mechanism and the subsequent pest control and prevention from the perspective of female sex pheromones and symbiotic fungi, which could lead to the development of improved attractants via the principle of chemical ecology.

**Supplementary Materials:** The following are available online at https://www.mdpi.com/article/10.3390/agronomy11122461/s1: Figure S1. Ramachandran plot, ERRAT, and Verify_3D result of SnocOBP7 model; Figure S2. Predicted binding pocket of SnocOBP7; Figure S3. The root-mean-square fluctuation (RMSF) curve of the 11 systems; Table S1. Cluster analysis of the SnocOBP7–ligand complexes based on the MD simulation trajectories.

**Author Contributions:** Conceptualization, E.-H.H. conceptualized and proposed the study idea; investigation, Y.-N.L. carried out the required biochemical experiments with the help of H.L.; Y.-N.L. and E.-H.H. carried out the bioinformatics, molecular docking, and molecular dynamics experiments; methodology, E.-H.H. and P.-F.L.; formal analysis, Y.-N.L. and H.L.; resources, P.-F.L., H.-L.Q. and X.-H.Y.; software, E.-H.H. and X.-H.Y.; writing—original draft preparation, Y.-N.L. and E.-H.H.; writing—review and editing, Y.-N.L., P.-F.L. and H.-L.Q. All authors have read and agreed to the published version of the manuscript.

**Funding:** This research was supported by the National Natural Science Foundation of China (Grant No. 31570643, 81774015) and the National Key R&D Program of China (2017YFD0600103). The funders had no role in study design, data collection and analysis, decision to publish, or preparation of the manuscript.

**Data Availability Statement:** The authors confirm that the data supporting the findings of this study are available within the article and its Supplementary Materials.

**Acknowledgments:** We thank anonymous reviewers and editors for their comments and suggestions on the manuscript.

**Conflicts of Interest:** The authors declare no conflict of interest.

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
