# Peer review of "Computational Interaction Analysis of Sirex noctilio Odorant-Binding Protein (SnocOBP7) Combined with Female Sex Pheromones and Symbiotic Fungal Volatiles"

_agronomy, doi:10.3390/agronomy11122461_

Round 1

Reviewer 1 Report

The manuscript submitted by Li et al, titled "Computational interaction analysis of Sirex noctilio odorant binding protein (SnocOBP7) combined with female sex pheromones and symbiotic fungal volatiles” aims to identify the protein binding sites for SnocOBP7. Overall, it is nicely written and the method is clean.

Minor comments

Line 16- “effective methods are unavailable” for what needs to be mentioned.

Line 86- “mainly including Pinus.” Remove including

The manuscript in general is too long- needs to be cut down.

male wood wasps should be in the parentheses at the first appearance of the scientific name.

Figures 4 and 7- enlarge x-, y-axis and figure legends.

Figure 5- mark hydrophobic residues

Reviewer 2 Report

This manuscript is well written, the research is interesting, the information expands our knowledge of S. noctilio biology, and should be published.  However, the authors should try to better explain how attractants detected by receptors in genitalia will be useful for managing S. noctilio.  Such receptors will not be useful monitoring populations because they are contact receptors and effective at only close range. I’m not saying the information can’t be useful, just that the authors’ have not properly expressed how it might be used.

Line 16: ‘effective methods are unavailable’, effective methods for what?

Line 17: how is the protein SnocOBP7 related to olfactory interactions, does it occur on the antennae?  If so, this information would be useful to the reader.

Line 34: Olfaction is one of the primary senses insects use to perceive their environment, not the primary sense for all insects as your sentence suggests.

Line 37: volatiles detected by an insect’s olfactory system may induce behavioral responses, they don’t ‘cause’ them.

Lines 62-65: It’s a big leap from chemical binding of OPBs and ligands to uncontrolled insect populations.

Lines 85-86: does S. noctilio have a ‘wide range of hosts’, are its host ‘primarily Pinus’, or are other evergreen genera included?

Lines 123-126: are these ‘male pheromones’ male produced aggregation pheromones that attract other males?

Line 139: if female sex pheromones are short distance contact pheromones, why use them in an attractant?  Contact pheromones would only work at close range, which is not much use in traps for monitoring an insect’s activity.

Lines 142-145: detection of pheromones by genitalia is not much use for long-distance detection.  How will this be used in detecting and managing S. noctilio?

Author Response

This manuscript is a resubmission of an earlier submission. The following is a list of the peer review reports and author responses from that submission.